# Continuous Compared to Accumulated Walking-Training on Physical Function and Health-Related Quality of Life in Sedentary Older Persons

**DOI:** 10.3390/ijerph17176060

**Published:** 2020-08-20

**Authors:** Pablo Monteagudo, Ainoa Roldán, Ana Cordellat, Mari Carmen Gómez-Cabrera, Cristina Blasco-Lafarga

**Affiliations:** 1Sport Performance and Physical Fitness Research Group (UIRFIDE), University of Valencia, 46010 Valencia, Spain; Ainoa.Roldan@uv.es (A.R.); Ana.Cordellat@uv.es (A.C.); 2Department of Education and Specific Didactics, Jaume I University, 12071 Castellon, Spain; 3Physical Education and Sports Department, University of Valencia, 46010 Valencia, Spain; 4Freshage Research Group, Department of Physiology, faculty of Medicine, University of Valencia, CIBERFES, Fundación Investigación Hospital Clínico Universitario/INCLIVA, 46010 Valencia, Spain; Carmen.Gomez@uv.es

**Keywords:** physical activity, public health, aging, dose-response, cardiorespiratory fitness, agility test, executive function, strength, older adults

## Abstract

The present study aimed to analyze the impact of overground walking interval training (WIT) in a group of sedentary older adults, comparing two different dose-distributions. In this quasi-experimental and longitudinal study, we recruited twenty-three sedentary older adults (71.00 ± 4.10 years) who were assigned to two groups of WIT. The continuous group (CWIT) trained for 60 min/session in the morning, while the accumulated group (AWIT) performed the same duration and intensity of exercise, but it was distributed twice a day (30 min in the morning and 30 more in the afternoon). After 15 weeks of an equal external-load training (3 days/week), Bonferroni post-hoc comparisons revealed significant (*p* < 0.050) and similar large improvements in both groups in cardiorespiratory fitness and lower limb strength; even larger gains in preferred walking speed and instrumental daily life activity, which was slightly superior for CWIT; and improvements in agility, which were moderate for CWIT and large for AWIT. However, none of the training protocols had an impact on the executive function in the individuals, and only the AWIT group improved health-related quality of life. Although both training protocols induced a general significant improvement in physical function in older adults, our results showed that the accumulative strategy should be recommended when health-related quality of life is the main target, and the continuous strategy should be recommended when weakness may be a threat in the short or medium term.

## 1. Introduction

Increasing evidence suggests that prolonged sedentary bouts [1,2] and/or few breaks in sedentary time [3,4] may damage metabolic health, independently of total sedentary time and moderate–vigorous physical activity (PA) [5]. High volumes of accumulated sedentary time contribute to increase the risk of cardiovascular disease, certain cancers, and premature mortality [6]. It is also associated with chronic, sterile, low-grade inflammation underlying the pathogenesis of many age-related diseases, described as inflammaging in older adults [7]. Since PA diminishes with age [8], sedentary time has become a world leading cause of death and disability according to the World Health Organization [9]. Similarly, research evidence confirms the negative impact of sedentary behaviors on physical fitness, especially in older people [10,11]. This is associated with the development of functional limitations, independently of PA levels [12]. Sedentary behaviors represent thus a serious public health problem in these older individuals.

Conversely, regular PA positively influences almost every human’s physiologic system or psychological aspect [13,14,15]. Exercise training is one of the most effective strategies for decreasing the likelihood of sedentary lifestyles and age-related diseases, thereby promoting independence in daily life activities and enhancing the quality of life in the growing older adult populations of many countries [16]. In this scenario, walking or brisk walking is the main example of moderate-intensity activity recommended by public health guidelines [17], because it is most frequently chosen by seniors [18], it requires minimal equipment, and it offers opportunities for friendship and social support [19]. Walking interventions, apart from being effective in increasing PA [20], are related to the prevention of cognitive decline [21] and to the improvement of health-related quality of life in older people [22]. Moreover, this kind of exercise may be a convenient activity to circumvent barriers to training, since it can be performed in the individual’s proximity zone, and it can be performed at a variety of intensities or modalities, either alone or in a group [23].

Classic proposals in walking programs to improve physical fitness in older persons have been established on the following basis: moderate intensity (5–6 on a 10 point scale where all-out effort is valued with a 10), minimum of 30 min, five days a week [24]. However, experimental findings have demonstrated that moderate-to-vigorous PA, accumulated in short bouts (>10 min) and totaling at least 30 min in duration, may be as effective as longer bouts in improving some disease risk factors, like plasma lipid profile, fasting plasma insulin levels, or body composition [25,26]. As a consequence, many PA guidelines have evolved to incorporate this recommendation [17,27]. In addition, some authors have proposed that PA guidelines in older adults should take into account the physical–behavioral binomial [28], applying the training principles based on people’s activity or sedentary patterns. In this way, exercise interventions in senior individuals should maintain active morning bouts and reduce sedentary behaviors in the afternoon and evening hours [29]. Accumulative exercise (distributing exercise training in the morning and in the afternoon) could be a good strategy to break and reduce sedentary time, especially in the less active time-slots. Furthermore, to our knowledge, no-studies have investigated whether accumulative proposals convey some advantage compared to similar doses of continuous exercise regarding physical function in sedentary older people. In addition, it remains unknown whether accumulated training enhances, more than does continuous training, the benefits in functional outcomes or health-related quality of life.

The aim of the present study was to analyze the impact of an overground walking program on physical function and health-related quality of life in a group of sedentary older adults comparing two different dose-distributions (accumulative versus continuous). Given the analyzed changes in body composition [25], we hypothesize that both strategies, tailored and periodized regarding their capacities, will be effective with differences in effects size.

## 2. Materials and Methods

### 2.1. Participants

Older adults from the Health Care Centre of Buñol (Manises Hospital area) were recruited to participate in the present study, which was approved by the ethics committee of the University of Valencia (H1484058781638). Inclusion criteria were as follows: ≥60 years old and fit to participate in a regular exercise program according to the medical referral; and sedentary (no participation in a regular exercise program or intentional activities beyond normal daily habits within the previous 4 months), and reporting a gait speed higher than 0.6 m/s. Exclusion criteria were presence of any disorder that would prevent the participant from being able to complete a training program, missing 4 or more consecutive training sessions, and adherence lower than 75% to the training sessions. A total of thirty-five older adults were screened, but only twenty-seven individuals met the inclusion criteria and signed the written informed consent. Participants were homogeneously stratified into 2 groups in terms of age, gender, body mass index (BMI), and gait speed in 6 m (this last categorized according to the “*Practical Guide for Prescribing a Multi-Component Physical Training Program to prevent weakness and falls in People over 70*”) [30]. Since three men and one woman did not complete the investigation for reasons not related to the study, 23 participants (*N* = 23; 71.0 ± 4.1 years; 75.6 ± 13.1 kg; 10 female) were included in the statistical analyses.

### 2.2. Research Design

The participants were enrolled to participate in this quasi-experimental and longitudinal study in January 2017. During February 2017, a baseline multidisciplinary team assessment of each participant was performed, comprising the gathering of relevant demographic and biological information, functional ability, health-related quality of life, executive function, and instrumental activities of daily living. Participants were assigned to two groups of supervised and tailored walking interval training (WIT) for 15 weeks. The continuous walking interval training (CWIT) group trained for 60 min/session, always in the morning, while the accumulated walking interval training (AWIT) group performed exactly the same duration and intensity of exercise, but distributed twice a day (30 min in the morning and 30 more in the afternoon) with at least 5-h separating each exercise bout. WIT programs started in March 2017. After the 15 weeks had ended (June 2017), all the participants were re-assessed by repeating the initial protocol.

### 2.3. Walking Interval Program (WIT)

All participants in the WIT program trained 3 times a week for 15 weeks (divided in 7 + 7 with a week of rest between weeks 7 and 8). Sets, repetitions, intensity, duration of work, and active recovery intervals are reported in Table 1. As previously described by our research group [25], intervals and intensities were increased and scheduled considering the rating of perceived effort (RPE 1–10) and adjusted by heart rate (HR) monitoring. Participants were instructed to walk close to the programmed RPE, reinforcing the first sessions with some RPE familiarization tasks. In order to control their HR (target: ≤80% HR_max_), they were also provided with a Beurer PM-15 HR monitor, without chest strap (Beurer, Ulm, Germany), and one individualized card with the HR estimated for every RPE zone. All sessions began with a brief warm-up period and ended with a cool down including breathing, stability, and joints mobility exercises.

### 2.4. Outcomes

We assessed different functional ability and psychosocial parameters before and after the intervention by the following tests and questionnaires.

#### 2.4.1. Grip Strength (GS)

Grip strength (GS) was evaluated by the Takei 5401 adaptable dynamometer (Takei Scientific Instruments CO., LTD, Tokyo, Japan). Following previous protocol [31], the contraction was maintained for 5 s, with the arm stretched along the body. Two measurements were taken on each side, with 1-min rest between them, and the best value was considered for the final analysis.

#### 2.4.2. Six Minute Walk Test (6MWT)

Cardiorespiratory fitness was evaluated with the six minute walk test (6MWT), according to the standard protocol [32], in a walking course of 30 m. Participants walked as fast as possible for 6 min, without running, being encouraged in each lap. They were warned of the time at 3 and 5 min.

#### 2.4.3. Five Times Sit-To-Stand Test (FTSST)

The five times sit-to-stand test (FTSST) was used to assess lower limb strength [33,34]. To perform the test, the subject was instructed to cross both arms across his/her chest and stand up completely, then sit and stand up a total of five times as quickly and safely as possible. Video recordings of the test were analyzed with the sports analysis video player software Kinovea (https://www.kinovea.org). Timing began when the subject’s buttocks took off from the seat and stopped when they returned to the seat after the fifth repetition. The test was performed once. If the subject did not perform part of the test correctly, he/she was stopped immediately by the investigator, and the test was restarted. The subject was closely guarded by the investigator to ensure the correct performance of the test and to prevent any injurious events.

#### 2.4.4. Preferred Walking Speed (PWS)

Preferred walking speed (PWS), also known as “most comfortable” or “self-paced” walking speed, was determined over ground on a 4.5 m walkway, using a system of two electric photocells by means of the Chronojump Software (Velleman PEM10D photocell, Cronojump Bosco System, response time 5–100 ms). Participants completed the distance (without acceleration but with a 2 m deceleration zone) walking at a comfortable and usual pace, and the mean of three attempts was taken as the PWS. This PWS was evaluated in the screening phase and after the intervention.

#### 2.4.5. Timed Up and Go Test (TUG)

We recorded the time that the participants took to rise from a chair, walk three meters, turn around, and walk back to the seated position. Every participant repeated the timed up and go test (TUG) three times and, when necessary, a rest period of up to 1 min was allowed in between tests. The stopwatch was started on the command “Go”, and the time was stopped when the test subject’s buttocks touched the chair seat again. The fastest of 3 timed trials were used for the reported testing. The fastest test was selected, and the participants did not receive verbal encouragement during the protocol [35,36].

#### 2.4.6. Executive Function

Executive function was assessed through the Stroop Color and Word Test [37]. This test consists of three parts that provide information on reading ability and psychomotor speed executive function, and that allow interference to be found in order to control the possible contaminating effect of the first 2 parts. Thus, the interference (IN) was used as a representative value of the executive function, according to the formula proposed by other authors [38].

#### 2.4.7. Instrumental Activities of Daily Living (IADL)

The VIDA questionnaire was used to assess instrumental activities of daily living (IADL). This questionnaire assesses the autonomous realization of 10 activities, using a Likert scale with 3–4 responses. The total summative score can range between 10 and 38 points. The VIDA questionnaire correlates well with the results of other tests assessing functioning like TUG and the Lawton and Brody scale [39,40].

#### 2.4.8. Health-Related Quality of life

The EQ-5D-5L was used to assess the health related quality of life [41]. This questionnaire has two parts: the EQindex, a descriptive profile that can be converted into an index-summary which defines health in terms of 5 dimensions (mobility, self-care, daily activities, pain/discomfort, and anxiety/depression); and the EQVAS, where respondents rate their overall health using a vertical visual analog scale from 0 to 100.

#### 2.4.9. Other Variables

The following parameters were also collected during the study: age, sex, weight, height, blood pressure, oxygen saturation, and HR. Arterial oxygen saturation (SpO_2_) and HR were determined with a pulse-oximeter attached to the fourth finger of the left hand (WristOx2-3150; Nonin, Plymouth, MN, USA), in a sitting position. Blood pressure was measured on the left arm with an Omron M3 Intellisense (HEM-7051-E) (Omron Healthcare, Kyoto, Japan) tensiometer. Standing height (m) was registered by means of a stadiometer (SECA 222, Hamburg, Germany). Subjects were measured without shoes with arms at their sides, looking straight, with knees together, and heels together. Shoulder blades, buttocks, and heels touched the measuring board. Measurement was taken at maximum inspiration with the head positioned in the Frankfort horizontal plane. After this, body weight (kg) was also registered by bioimpedance (TANITA, model BC-545N, Tokyo, Japan). Participants were weighed in light clothing controlling food intake in the previous hours to reproduce the evaluation conditions. BMI was calculated by dividing weight (kg) by height squared (m^2^).

### 2.5. Statistical Analysis

The analysis of the data was performed with the SPSS statistics package version 23 (IBM SPSS Statistics for Windows, Chicago, IL, USA). After testing for normality (Shapiro–Wilks), Student’s *t* test or the Mann–Whitney U test (SpO_2_) were first applied for baseline group comparisons. A repeated measures ANOVA was then conducted to analyze changes in health-related quality of life and functional measures, considering the main effect of the intervention (pre-post overall comparison) and the interaction between type*dose-distribution (CWIT vs. AWIT). Within-subjects effects tests at the first level, followed by Bonferroni post-hoc tests, were performed with statistical significance set at the level of *p* ≤ 0.05. Later on, in order to homogenize and analyze these changes, the effect size (ES) was calculated by means of the Cohen’s d, where the effect was considered small (d = 0.20–0.40), medium (d = 0.50–0.70), or large (d = 0.80–2.0) according to Cohen [42]. Descriptive statistics were expressed as mean ± standard deviation (SD).

Changes in functional and health-related quality of life variables were further expressed as percentage of change (calculated by means of the formula (post-score − pre-score)/pre-score × 100). Student’s t test or the Mann–Whitney U test were applied looking for group comparisons within deltas. Individual variables were checked for homogeneity of variance using Levene’s test. When differences were found in any functional variable, Spearman correlations were performed considering BMI to give light to these changes.

## 3. Results

Table 2 includes the baseline characteristics of the participants. The CWIT and AWIT groups were homogenous at the baseline features, with no statistically significant differences observed in terms of age, gender, weight, height, BMI, systolic blood pressure (SBP), diastolic blood pressure (DBP), SpO_2_, and HR. Functional and health-related quality of life outcomes were neither different at baseline (*p* > 0.05). Participants completed the intervention with an adherence rate of 83.75%.

The within-subjects effects test showed a significant main effect of “intervention” (*p* ≤ 0.05), but there was not effect of the interaction “intervention × dose-distribution” strategy for any variable. Only PWS, as revealed by a trend to significance (*p* < 0.100) (Table 3).

Regarding the main effect of the intervention, in the whole sample (Table 3), the repeated-measures ANOVA showed significant improvements in lower limb strength (FTSST: 11.77 ± 3.36 vs. 9.10 ± 1.34 s), cardiorespiratory fitness (6MWT: 522.26 ± 64.60 vs. 597.54 ± 80.28 m), PWS (1.16 ± 0.19 vs. 1.40 ± 0.15 m/s), agility (TUG: 7.40 ± 1.28 vs. 6.60 ± 0.89 s), autonomy (IADL: 32.83 ± 2.64 vs. 36.09 ± 1.76), health-related quality of life (EQindex: 0.85 ± 0.14 vs. 0.91 ± 0.09), and BMI (29.11 ± 3.99 vs. 28.45 ± 3.86). Significant decrements were found in GS (34.65 ± 9.55 vs. 32.72 ± 10.21 kg). Non-significant changes were found for executive function or EQVAS.

Further Bonferroni analyses of pre-post differences (Table 4) showed significant improvements for both strategies on FTSST, 6MWT, PWS, TUG and IADL with large and moderate ES. The 6MWT and FTSST showed similar large ES (d ≈ 1.0), while PWS and IADL showed large ES with a slight and superior difference for the CWIT group. Conversely, the AWIT group revealed a large ES for the TUG (d = 1.00), while the CWIT group presented a moderate ES (d = 0.49).

The EQindex showed a significant increase only for the AWIT group with a moderate ES (d = 0.66). No statistically significant variations were detected in any group for IN or EQVAS. Finally, GS showed a significant decrease in the AWIT group but with a small ES (d = 0.21).

The percentage of change in the main outcomes measured in our study are reported in Figure 1 (physical function parameters) and Figure 2 (executive function, IADL and health-related quality of life). We did not find statistically significant differences between CWIT and AWIT in any of the measurements performed. Only PWS showed a trend to significance (*p* = 0.097), with a higher change for the CWIT (27%) with regard the AWIT group (18%). Moreover, no-significant correlations (*p* > 0.05) were found between the percentage of change in functional outcomes and the BMI (deltas).

Finally, although we did not find significant differences for executive function, we observed that the CWIT group showed a negative percentage of change, while the AWIT group showed a positive one.

## 4. Discussion

The purpose of this study was to investigate the effects of continuous versus accumulated bouts of walking interval training on physical function and health-related quality of life in older persons. Our first hypothesis was that both strategies would be effective in the improvement of physical function and health related quality of life. In accordance with this hypothesis, our results showed that both interventions evoke similar benefits on physical function and daily life activities, so the accumulation of morning/afternoon exercise bouts does not exceed the beneficial effects of continuous walking training in sedentary older adults. However, PWS, IADL, and TUG reported different ES regarding both strategies. Additionally, we found no-improvements in GS, and only the accumulated strategy was able to improve the health related quality of life after 15 weeks of walking training.

The effect of starting physical exercise on physical function seems evident in sedentary older adults when it is properly programmed, tailored, and supervised by professionals [43,44], even at very advanced ages [45]. Despite the medium duration of the intervention (15 weeks) [46], we found similar improvements in cardiorespiratory fitness and lower limb strength, independently of the protocol. Enhancing these variables over 15–20%, with a large ES (whatever the group and outcome) is very important for older adults, since lower limb strength is related to falls prevention [47], and cardiorespiratory fitness is an independent risk factor in terms of mortality from any cause [48].

Previous studies have evaluated the influence of continuous versus accumulated exercise during the day, especially in cardiorespiratory fitness, with similar results for both strategies [49,50,51]. Nevertheless, these studies were conducted with younger subjects (<65 years) and/or with specific pathologies (obese, diabetic, etc.) [49,50,51]. In addition, the duration of the exercise bouts may vary among different studies between 10 and 30 min. For instance, Karstoft et al. [52] found that both the accumulated walking bouts group (3 × 10 min/day) and continuous walking group (30 min/day) significantly increased the aerobic condition after eight weeks. In the same way, other authors found improvements in both strategies in obese sedentary adults (continuous group achieved increases of 8% and accumulated group increased by 6%) in cardiorespiratory fitness [53]. Whatever the strategy, the improvements were lower than the ones found in our study (about 15%). However, some studies have shown slight differences, with the continuous strategy showing superior results in the first six months training [54], which were equated with the accumulated strategy at 18 months. These data suggest that to induce short-term changes, continuous strategies might be more effective.

The gradual increase in exercise-intensity and the modulation of the lap-duration through the whole WIT, with walking intensities close to the anaerobic threshold in the last weeks of the intervention (RPE of 7), can explain some of these differences. In the same way, as the walking speed increases during exercise, the length of the step and the amplitude of the movement also increase, which is an extra stimulus for the neuromuscular system [55]. Hence, the intensity of exercise in older adults is an important matter [56,57,58]. Previous studies have already theorized that both accumulated and continuous strategies could induce different adaptations due to differences in the effective intensity of exercise (internal load). Concentrating the load may lead to more pronounced physiological alterations [50], so the continuous dose would induce higher demands for exercises with similar volume and external intensity, or even RPE, improving cardiovascular responses.

Regarding the changes in agility, it was significantly improved for both doses of WIT without significance differences between groups, even though the ES was larger in the accumulated strategy when compared to the continuous one. The most time spent in a sedentary behavior may explain this result [59]. Agility usually comprises accelerations, decelerations, stop-and-go patterns, changes of direction (cutting maneuvers), and eccentric loads [60]. Therefore, it could be hypothesized that older adults who break sedentary behaviors more frequently could have slight benefits in this variable not only related to exercise training, but due to displacement and higher motor-time associated with the assistance to the training place.

Conversely, PWS and IADL showed a greater ES for the continuous strategy. Self-regulating walking speed over a longer consecutive training time (60 min vs. 30 min) may have resulted in participants in the continuous group being able to automate and apply that walking speed over a shorter distance (such as 6 m). This could explain the different ES in PWS (CWIT = 1.66 vs. AWIT = 1.24), which does not vary in aerobic capacity between strategies (CWIT = 1.02 vs. AWIT = 1.01). Regarding IADL, improvements in this outcome may be associated with greater physical function due to the increase in gait speed [61]. Therefore, our results support this association. In any case, improvements in these variables are very important in older adults. For example Perera et al. [62] highlighted that a higher walking speed is associated with a lower incidence of disability or mobility problems in three years; and the ability to perform complex tasks that involve IADL (such as the responsibility for preparation and taking medication, the control of domestic economy, the use of transport or telephone, etc.) are basic in older adult lives and tend to decline quickly [63]. Unfortunately, the sample size was not enough to consider any possible influence of gender in this different response to exercise training. However, according to Scaglioni et al. [64], there are no gender differences in the older adults’ gait speed, despite the fact that several domains of this gait are sex-related, and hence sex differences should be considered when preparing exercise programs in older adults.

As mentioned above, no dose-distribution has managed to produce positive effects on GS. In fact, some authors have already pointed out that in programs aimed at aerobic conditioning, improvements in strength depend largely on the muscle group involved in this type of exercise [44]. Specifically, our data reflect a slight and significant decrease in GS in the accumulated group (low ES). Loss of GS has been shown to vary according to BMI trajectory [65], which was significant for AWIT, and just a trend for CWIT, but we have found no-correlation between the percentage of change in GS and BMI in our study. Further studies will elucidate if 60 min are always enough stimuli to ensure the reaching of the minimum neuromuscular (or metabolic) demands to maintain upper limb strength, since accumulating this duration does not prevent loses in this capacity. Consistent with these results, Rooks et al. [66] found that an intervention of continuous walks (45 min) at an intensity chosen by the participants maintained the levels of GS compared to the controls.

Although positive effects of aerobic training programs have been demonstrated to elicit selective impacts in areas of executive functions (such as multi-tasking, planning, and inhibition) [11,12,13], surprisingly, in this study we found no-significant improvements in executive function (interference) after the intervention. Recently, it has been observed how cognitive performance (inhibition function) could be maintained or even improved after a period of training cessation [67], experimenting with a slower and more delayed evolution regarding physical function [68,69]. Given that detraining has not been evaluated in this study, we cannot confirm this effect, but in any case, some authors consider that maintaining executive function at these ages can also be considered a beneficial effect after this type of intervention [70]. In addition, we point out the different percentage of change for each exercise group: negative for continuous group, and positive for the accumulated (but without significant differences). The high dispersion in this variable can provide an idea of the heterogeneity of older adults and suggest that WIT could have benefited some participants, so exercise-training individualization matters.

Our results regarding health-related quality of life show a significant improvement for the descriptive index of the Euroqol, but in the posterior analysis of Bonferroni comparisons, we found that this difference is due to the AWIT group. Several factors can influence the quality of life in older populations. Functional capacity, an active lifestyle, and good social relationships are some factors that explained subjective quality of life [71]. Some authors have identified that factors such as interaction with family and friends, enjoying nature, and being helpful to others are also important to the quality of life of older adults [72]. Therefore, it is possible that the accumulated groups, having to carry out a greater number of sessions, generate more opportunities for social encounters and mutual help before and after the exercise session (such as sharing private transport to go to the training sessions). Moreover, the walking program took place in a natural environment, with the psychosocial advantages that this entails [72,73].

The main limitation of our study is that we could not compare PA levels or changes in sedentary behavior after the intervention. Quantifying PA levels through accelerometry could have given more information about the transfer of both strategies towards more active lifestyles. In addition, the small size of the sample implicates some uncertainty when extrapolating the results and becomes another limitation of our study, jointly with the slightly different sexual composition between groups in the final sample (in the second group the males were twice as many as the females). Future research is needed to confirm that accumulated and continuous exercise produce similar but not equal improvements in older adults following walking interval training, and whether other types of exercise (like multicomponent programs) evoke differences in functional or quality of life outcomes.

## 5. Conclusions

As far as we know, this is the first study to compare the benefits of accumulating or concentrating the dose of a WIT on physical function and health-related quality of life in sedentary older persons. Public health policies aimed to prescribe walking exercise programs must account for the specific implications of exercise dose-distributions also in this population, knowing that exercise prescription is a complex process and requires manipulating individual training loads.

According to our results, when comparing the effect of splitting a continuous bout of interval-walking into shorter bouts (of equivalent total duration dispersed throughout the day), the accumulated program provides, at least, similar benefits on physical function compared to the continuous program. These findings provide further evidence that bout length is not a determinant of the health functional effects associated with exercise. Moreover, while continuous exercise can provide a slight difference for PWS, fractionalizing a single exercise into two series throughout the day can do so for autonomy, agility, and health-related quality of life. It is noteworthy that accumulated strategies may also have additional consequences, helping to change sedentary behavior in the short-term.

## Figures and Tables

**Figure 1 ijerph-17-06060-f001:**
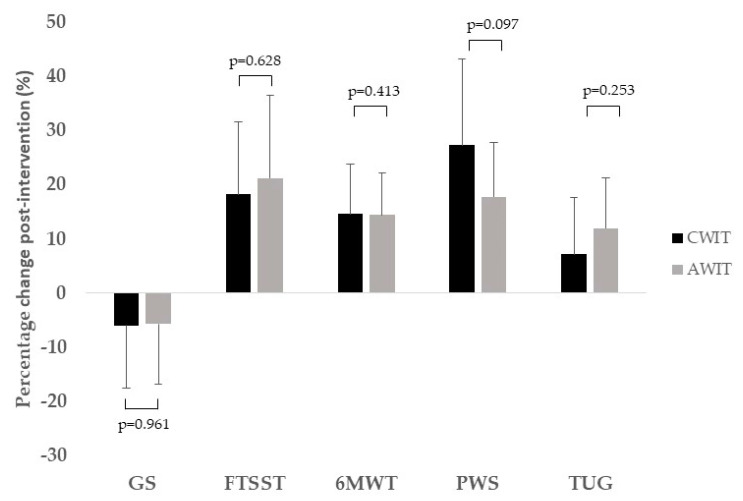
Percentage of change in physical function measurements following CWIT (solid bar) and AWIT (gray bar). GS indicates grip strength; FTSST, lower limb strength (time in seconds); 6MWT, 6 min walk test (m); PWS, preferred walking speed (time in seconds); TUG, timed up and go (time in seconds); CWIT, continuous walking interval training; AWIT, accumulated walking interval training.

**Figure 2 ijerph-17-06060-f002:**
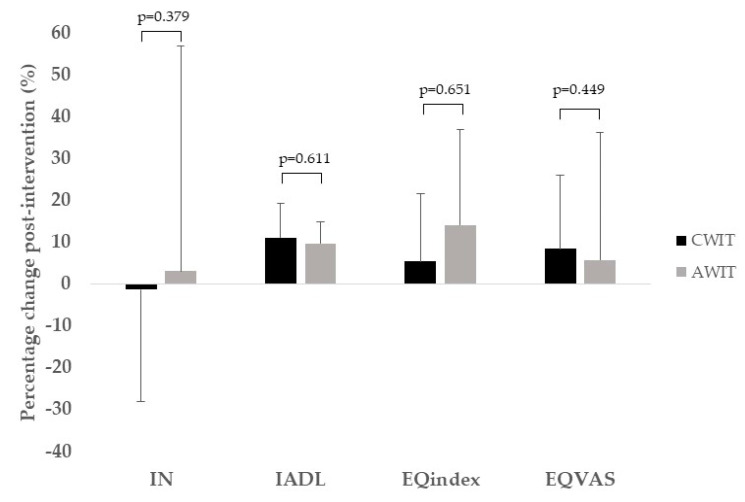
Percentage of change in executive function, instrumental activities of daily living, and health-related quality of life measurements following CWIT (solid bar) and AWIT (gray bar). IN, inhibition; IADL, instrumental activities of daily living; CWIT, continuous walking interval training; AWIT, accumulated walking interval training.

**Table 1 ijerph-17-06060-t001:** Walking Interval Training.

	Session	Session Description (Sets × Repetitions (Work + Recovery))	Total Session Duration (min)		Session	Session Description (Sets × Repetitions (Work + Recovery))	Total Session Duration (min)
Week 1	1	* 2 × 5 (2 min 4 RPE + 2 min 2 RPE)	40	Week 9	20	1 × 7 (2 min 6 RPE + 4 min 4 RPE)	42
	2	1 × 10 (2 min 4 RPE + 2 min 2 RPE)	40		21	1 × 10 (2 min 6 RPE + 2 min 4 RPE)	40
	3	1 × 10 (2 min 4 RPE + 1.5 min 2 RPE)	35	Week 10	22	1 × 5 (2 min 7 RPE + 3 min 4 RPE)	45
Week 2	4	1 × 5 (4 min 4 RPE + 4 min 2 RPE)	40		23	1 × 5 (2 min 7 RPE + 3 min 4 RPE)	40
	5	1 × 5 (4 min 4 RPE + 4 min 3 RPE)	35		24	1 × 5 (3 min 6 RPE + 3 min 4 RPE)	37.5
	6	1 × 7 (4 min 4 RPE + 4 min 2 RPE)	42	Week 11	25	1 × 6 (2 min 7 RPE + 1.5 min 4 RPE)	42
Week 3	7	* 2 × 5 (2 min 5 RPE + 2 min 3 RPE)	40		26	1 × 7 (3 min 7 RPE + 3 min 7 RPE)	42
	8	1 × 10 (2 min 5 RPE + 2 min 3 RPE)	40		27	1 × 10 (2 min 7 RPE + 1 min 5 RPE)	40
Week 4	9	1 × 5 (4 min 5 RPE + 4 min 3 RPE)	40	Week 12	28	1 × 6 (4 min 6 RPE + 2 min 4 RPE)	42
	10	1 × 6 (4 min 5 RPE + 3 min 3 RPE)	42		29	1 × 7 (2 min 7 RPE + 1 min 5 RPE)	42
	11	1 × 8 (4 min 5 RPE + 2 min 3 RPE)	48		30	1 × 8 (2.5 min 7 RPE + 3 min 4 RPE)	44
Week 5	12	1 × 10 (2 min 6 RPE + 2 min 4 RPE)	40	Week 13	31	1 × 5 (4 min 6 RPE + 1.5 min 5 RPE)	45
	13	1 × 10 (2 min 6 RPE + 2 min 4 RPE)	40		32	1 × 5 (3 min 6 RPE + 3 min 4 RPE)	40
Week 6	14	1 × 5 (4 min 6 RPE + 4 min 4 RPE)	40	Week 14	33	1 × 6 (2 min 7 RPE + 1.5 min 4 RPE)	42
	15	1 × 6 (4 min 6 RPE + 3 min 4 RPE)	42		34	1 × 5 (3 min 7 RPE + 3 min 7 RPE)	40
	16	1 × 8 (4 min 6 RPE + 2 min 4 RPE)	48		35	1 × 6 (3 min 7 RPE + 1 min 5 RPE)	42
Week 7	17	1 × 9 (2 min 7 RPE + 2 min 4 RPE)	36	Week 15	36	1 × 7 (4 min 7 RPE + 3 min 5 RPE)	42
	18	1 × 10 (2 min 7 RPE + 2 min 4 RPE)	40		37	1 × 7 (3 min 7 RPE + 2.5 min 5 RPE)	38.5
	19	1 × 10 (2 min 7 RPE + 2 min 4 RPE)	40		38	1 × 8 (2 min 7 RPE + 2 min 5 RPE)	40

* When two blocks were performed, 1 to 3 min of break were considered in order to drink water and rest. RPE: the rating of perceived effort.

**Table 2 ijerph-17-06060-t002:** Physical characteristics at baseline.

	Total, *N* = 23	CWIT, *N* = 11	AWIT, *N* = 12
Age, years	71.0 ± 4.1	71.7 ± 3.3	70.3 ± 4.7
Weight, kg	75.9 ± 13.2	71.5 ± 11.1	79.9 ± 14.2
Height, m	1.6 ± 0.1	1.6 ± 0.1	1.6 ± 0.1
BMI, kg/m^2^	29.1 ± 4.0	27.8 ± 3.1	30.3 ± 4.5
SBP, mmHg	152.4 ± 16.0	151.4 ± 14.4	153.4 ± 17.9
DBP, mmHg	82.5 ± 10.5	83.4 ± 9.5	81.7 ± 11.7
SpO_2_, %	94.4 ± 4.5	95.3 ± 2.6	93.6 ± 5.7
HR, bpm	73.4 ± 10.7	75.2 ± 12.4	71.7 ± 9.0
Gender			
Females, % (*n*)	39.1 (9)	45.5 (5)	33.3 (4)
Males, % (*n*)	60.9 (14)	54.5 (6)	66.7 (8)

BMI: body mass index; SBP: systolic blood pressure; DBP: diastolic blood pressure; SpO_2_: arterial oxygen saturation; HR: heart rate. CWIT: Continuous Walking Interval Training; AWIT: Accumulated Walking Interval Training.

**Table 3 ijerph-17-06060-t003:** Tests of within-subjects’ effects.

	Variable	Type III Sum of Squares	df	Mean Square	F	*p*	Partial Eta Squared
Intervention	BMI	4.906	1	4.906	7.875	0.011 *	0.273
	GS	42.13	1	42.13	6.72	0.017 *	0.243
	FTSST	80.29	1	80.29	21.07	0.001 **	0.501
	6MWT	65,140.41	1	65,140.41	78.48	0.001 **	0.789
	PWS	0.68	1	0.68	90.58	0.001 **	0.812
	TUG	7.22	1	7.22	20.17	0.001 **	0.490
	IN	1.69	1	1.69	0.04	0.836	0.002
	IADL	122.66	1	122.66	66.35	0.001 **	0.760
	EQindex	0.04	1	0.04	4.33	0.050 *	0.171
	EQVAS	145.52	1	145.52	1.44	0.243	0.064
Intervention × Dose-distribution	BMI	0.045	1	0.045	0.073	0.790	0.003
	GS	0.63	1	0.63	0.10	0.755	0.005
	FTSST	2.85	1	2.85	0.75	0.397	0.034
	6MWT	15.62	1	15.62	0.02	0.892	0.001
	PWS	0.03	1	0.03	3.63	0.070 ^ƚ^	0.147
	TUG	0.47	1	0.47	1.32	0.263	0.059
	IN	1.99	1	1.99	0.05	0.822	0.002
	IADL	0.39	1	0.39	0.21	0.648	0.010
	EQindex	0.01	1	0.01	0.78	0.387	0.036
	EQVAS	41.17	1	41.17	0.41	0.530	0.019

BMI: body mass index; GS: grip strength; FTSST: five times sit to stand test; 6MWT: 6 min walk test; PWS: preferred walking speed; TUG: timed up and go; IN: interference; IADL: instrumental activities of daily living; EQindex: descriptive index of Euroqol; EQVAS: visual analogue scale of Euroqol. ** *p* ≤ 0.001; * *p* ≤ 0.050; ^ƚ^
*p* ≤ 0.100.

**Table 4 ijerph-17-06060-t004:** Measures for continuous (*n* = 11) and accumulated groups (*n* = 12).

	Pre-CWIT	Post-CWIT	ES	Pre-AWIT	Post-AWIT	ES
BMI, kg/m^2^	27.8 ± 3.1	27.3 ± 3.1 ^ƚ^	0.2	30.3 ± 4.5	29.5 ± 4.3 *	0.2
GS, kg	33.1 ± 9.1	31.4 ± 10.1	0.2	36.1 ± 10.1	33.93 ± 10.6 *	0.2
FTSST, s	10.9 ± 2.2	8.7 ± 1.5 *	1.1	12.6 ± 4.1	9.5 ± 1.2 **	1.0
6MWT, m	529.9 ± 68.5	606.4 ± 80.6 **	1.0	515.2 ± 63.0	589.4 ± 82.6 **	1.0
PWS, m/s	1.1 ± 0.2	1.4 ± 0.2 **	1.7	1.2 ± 0.2	1.4 ± 0.1 **	1.2
TUG, s	7.1 ± 1.4	6.5 ± 1.0 *	0.5	7.7 ± 1.2	6.7 ± 0.8 **	1.0
IN	−4.7 ± 8.7	−3.9 ± 10.3	0.1	−8.3 ± 5.7	−8.3 ± 6.5	0.0
IADL	32.5 ± 2.6	36.0 ± 1.6 **	1.6	33.1 ± 2.7	36.2 ± 1.9 **	1.3
EQindex	0.9 ± 0.1	0.9 ± 0.1	0.3	0.8 ± 0.2	0.9 ± 0.1 *	0.7
EQVAS	74.1 ± 17.1	79.6 ± 18.1	0.3	77.5 ± 14.8	79.2 ± 16.0	0.1

BMI: body mass index; GS: grip strength; FTSST: five times sit to stand test; 6 MWT: 6 min walk test; PWS: preferred walking speed; TUG: timed up and go; IN: interference; IADL: instrumental activities of daily living; EQindex: descriptive index of Euroqol; EQVAS: visual analogue scale of Euroqol; ES: effect size. ** *p* ≤ 0.001; * *p* ≤ 0.050; ^ƚ^
*p* ≤ 0.100.

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
