# Peer review of "Continuous Compared to Accumulated Walking-Training on Physical Function and Health-Related Quality of Life in Sedentary Older Persons"

_ijerph, 2020, doi:10.3390/ijerph17176060_

Round 1

Reviewer 1 Report

This paper presents new interesting data on the effects of different  PA dose-distributions in  sedentary older people, analyzing the impact of an overground walking program  on physical  function  and  health-related  quality  of  life. Although the sample size was small, the paper is well presented and I recommend it for publication after the following points have been addressed:

Materials and Methods

Line 91: what was the size of the group from which the 27 subjects were extracted? 

Line 93: indicate the unit of measurement in full: 6 meters.

Line 96: replace semicolon with a comma: “related to the study, 23 participants”

Lines 96-97: correct the numbers rounding to one decimal place.

Line 100: change “a baseline multidisciplinary assessment” with “a baseline multidisciplinary team assessment”

Paragraph 2.4.9: The description of anthropometric techniques is too brief. Specify at least the position of the head (plane) for the measurement of height and references.

Line 193: change “Descriptives” with “Descriptive statistics”.

Results

Table 2:

  • add the unit of measurement of the age variable
  • correct all the numbers rounding to one decimal place

Table 3: why did not use a repeated-measures t-test?

Lines 228-230: enter p values in a single line in the footnote.

Table 4: correct all the numbers rounding to one decimal place.

Lines 245-247: enter p values in a single line in the footnote.

Discussion

The two groups compared have a different sexual composition (in the second group the males are twice as many as the females), how could this have affected the trend detected? Certainly it is necessary to underline this relevant aspect among the limitations of the study.

Reviewer 2 Report

The paper is original, well written, the statistical analyses is appropriate, the results are clearly described but the conclusions need to be revised.

A revision is due:

To our opinion the conclusion in that "AWIT group diminished significantly grip strength, but improved in health-related  quality of life" was not supported by experimental approach. In fact, the author's  failed to evaluate the strength of the upper limbs, further a randomized cross-over design seems most appropriate in order support the this conclusion.We suggest to the authors to revise the manuscrispt as above indicated and discuss only on the QoL markers.

Reviewer 3 Report

Dear editor and authors, thank you very much for the opportunity to review this manuscript. It is a novel and interesting study about physical exercise interventions to older adults considering two different dose-distributions. I have some minor corrections to suggest:

1 – Although still used, the term elderly seems to be outdated, please prefer to use “older adults”, “older men”, “older women”, “older persons” instead of elderly (refer to United Nations recommendations).

2 - Please carefully verify (throughout the text) if objective-methods-results verbs are in past tense. Abstract: Line 18 – “aims” change to “aimed”; Line 29 – “induce” change to “induced”; Line 30 – “show” change to “showed”;

3 – Line 27 – “Old individuals” change to “individuals”;

4 – In all the manuscript, please consider to follow this convention: all abbreviations/acronyms should be determined at the first time in which the full term appear. After that, please use only the abbreviation/acronym. In some cases – when using OA to older adults, for example - several synonyms are used and the acronym is unnecessary. Also, it seems unusual to specify acronym/abbreviation for the studied population/sample. Please keep acronyms/abbreviations for institutions and tests/outcomes. Finally, please check the text and consider to reduce the number of acronyms.

5 – Line 77- “The aim of the present study is to…” change to “The aim of the present study was to…”;

6 - Line 118 – Instead of “Borg zone” please use “RPE zone”;

Some examples to be correct in all sections about acronym/abbreviations: Line 39 – “moderate-vigorous physical activity” change to “moderate-vigorous physical activity (PA)”; Line 42 – “Since physical activity” change to “Since PA”; Line 46 – “physical activity (PA)” change to “PA”; Line 84 – “Older adults” change to “OA”; Line 93 – “BMI” change to “body mass index (BMI)”; Line 116 – “heart rate” change to “HR”; Line 126 – “Grip strength (GS)” change to “GS”; Line 176 – “heart rate” change to “HR”; Line 176 – “Heart Rate (HR)” change to “HR”; Line 181 – “Body Mass Index (BMI)” change to “BMI”; Line 194 – “SD” change to “standard deviation (SD)”; Line 204 – “SBP” change to “systolic blood pressure (SBP)” ; “DBP” change to “diastolic blood pressure (DBP)”; Line 216-217 – “preferred walking speed” change to “PWS”; Line 218-219 – “body mass index” change to “BMI”; Line 220 – “grip strength”  change to “GS”; Line 232 to Line 238 – change all “effect sizes” to “ES”; Line 275 – “older adults” change to “OA”; “effect sizes” to “ES”; Line 277-278 – “older adults” change to “OA”; Line 281 – “effect size” change to “ES”; Line 282 – “older adults” to “OA”; Line 302-303 – “older adults” to “OA”; Line 309 – “effect size” to “ES”; Line 313 – “older adults” to “OA”; Line 316 – “effect size” to “ES”; Line 319 – “effect size” to “ES”; Line 323 – “older adults” to “OA”; Line 329-330 – “grip strength” to “GS”; Line 333 – “effect size” to “ES”; “grip strength” to “GS”; Line 334- “grip strength” to “GS”; Line 339 – “grip strength” to “GS”; Line 350-351 – “older adults” to “OA”; Line 364 – “physical activity” to “PA”; Line 365 – “physical activity” to “PA”; Line 383 – “preferred walking speed” to “PWS”; “grip strength” to “GS”;

These are just examples, please carefully check the text. Following previous suggestion, only change “older adults” to “OA” if you insist in the use of acronym in this case. Otherwise, reconsider use preferred terms “older adults, older people, older persons, older men, older women…” instead of the acronym “OA”;

I have no comments about the methodology of this study. 
